# Bone Marrow-Derived Mesenchymal Stem Cell Implants for the Treatment of Focal Chondral Defects of the Knee in Animal Models: A Systematic Review and Meta-Analysis

**DOI:** 10.3390/ijms24043227

**Published:** 2023-02-06

**Authors:** Ernest Lee, Ilias Ektor Epanomeritakis, Victor Lu, Wasim Khan

**Affiliations:** 1Ipswich Hospital, East Suffolk and North Essex NHS Foundation Trust, Ipswich IP4 5PD, UK; 2Bedford Hospital South Wing, Bedfordshire Hospitals NHS Foundation Trust, Bedford MK42 9DJ, UK; 3School of Clinical Medicine, University of Cambridge, Cambridge CB2 0SP, UK; 4Department of Trauma and Orthopaedic Surgery, Addenbrooke’s Hospital, University of Cambridge, Cambridge CB2 0QQ, UK

**Keywords:** cartilage, regeneration, bone marrow-derived mesenchymal stem cells, implant, scaffold, cartilage to cartilage integration

## Abstract

Osteoarthritis remains an unfortunate long-term consequence of focal cartilage defects of the knee. Associated with functional loss and pain, it has necessitated the exploration of new therapies to regenerate cartilage before significant deterioration and subsequent joint replacement take place. Recent studies have investigated a multitude of mesenchymal stem cell (MSC) sources and polymer scaffold compositions. It is uncertain how different combinations affect the extent of integration of native and implant cartilage and the quality of new cartilage formed. Implants seeded with bone marrow-derived MSCs (BMSCs) have demonstrated promising results in restoring these defects, largely through in vitro and animal studies. A PRISMA systematic review and meta-analysis was conducted using five databases (PubMed, MEDLINE, EMBASE, Web of Science, and CINAHL) to identify studies using BMSC-seeded implants in animal models of focal cartilage defects of the knee. Quantitative results from the histological assessment of integration quality were extracted. Repair cartilage morphology and staining characteristics were also recorded. Meta-analysis demonstrated that high-quality integration was achieved, exceeding that of cell-free comparators and control groups. This was associated with repair tissue morphology and staining properties which resembled those of native cartilage. Subgroup analysis showed better integration outcomes for studies using poly-glycolic acid-based scaffolds. In conclusion, BMSC-seeded implants represent promising strategies for the advancement of focal cartilage defect repair. While a greater number of studies treating human patients is necessary to realize the full clinical potential of BMSC therapy, high-quality integration scores suggest that these implants could generate repair cartilage of substantial longevity.

## 1. Introduction

Osteoarthritis is a chronic disease, primarily affecting the elderly or obese population, involving diffuse degeneration of articular cartilage in load-bearing joints [1,2]. Acquiring focal cartilage defects through traumatic injuries is another risk factor for developing the condition, one which commonly plagues young, active patients. While a minority of cartilage defects may regress, most increase in size in otherwise healthy people [3]. Given that current, less invasive treatment options cannot ultimately prevent the need for joint replacement [4], these injuries pose the risk of requiring early joint replacement and multiple subsequent revision surgeries in young patients [5].

Given the significant individual and organizational costs posed by knee arthroplasty [6] and the projected increase in the incidence of revision surgeries [7], a method capable of preventing the progression of focal defects would be invaluable. Cell therapies have emerged as a potential solution to the repair of cartilage tissue, which is renowned for its poor regenerative capability [8]. Matrix-induced autologous chondrocyte implantation (MACI) has proven efficacious compared to other surgical techniques, such as microfracture and mosaicplasty, and is approved for use in human patients in clinical settings [9,10].

The MACI technique, while effective, does pose a number of limitations. These may be overcome by mesenchymal stem cells (MSCs). Reduced cost, better capability of MSCs to proliferate in vitro, and their potentially greater tendency to produce hyaline rather than fibrocartilage all make MSC therapy an attractive future option [8,11,12,13]. Its one-step operation also reduces the risk of the donor–site morbidity inherent in the MACI process, in which cartilage harvest precedes chondrocyte implantation at a separate site. Despite this potential, the use of MSCs remains a largely experimental therapy, with limited application in humans as of yet [8].

Indeed, our previous systematic review investigating the integration of cell-loaded implants with native cartilage in humans identified a paucity in studies using MSC therapy [4]. Therefore, we aimed to gain a comprehensive understanding of the ability of MSCs to generate repair cartilage, integrated with neighboring host cartilage, by compiling the results of animal studies in this PRISMA review. A broad range of permutations in MSC source and delivery methods have been trialed in multiple experimental models [14]. To ensure comparability between studies, we narrowed our focus to bone marrow-derived MSCs (BMSCs) which were surgically implanted directly onto cartilage defects within organic scaffolds or gels.

The integration of regenerated and native cartilage has proven difficult to achieve, and yet it is vital for the lifespan of repair [15]. This may be because the integration boundary remains the weakest point in the repair tissue, increasing the likelihood of tissue breakdown if integration is inadequate [16]. BMSCs have shown encouraging preliminary integration results in animals [17]. While the mechanism of cartilage–cartilage integration remains obscure, the paracrine effects of an extracellular matrix on chondrocytes and collagen cross-linking are considered relevant [13].

Also affecting the lifespan of repair is the morphology of the cartilage formed [18]. The tendency of early surgical techniques such as microfracture to produce fibrocartilage spurred the development of newer techniques which may produce repair tissue more akin to native hyaline cartilage. Fibrocartilage is rich in type I collagen, low in proteoglycan content, and contains scattered round cells, while hyaline cartilage is rich in both type II collagen and proteoglycan, with rounded chondrocytes divided into three zones of development [19]. Safranin O and Toluidine blue stains both stain specimens according to their proteoglycan content and are, therefore, commonly used in the assessment of cartilage repair quality [20].

As we found previously [4], human studies are unlikely to involve histological assessment of repair tissue. Instead, magnetic resonance imaging (MRI) is used as a high-quality, non-invasive surrogate technique to assess the morphology of the repair, including border integration [21]. As well as increasing our understanding of the potential for BMSC-seeded implants to yield integrated repair tissue, the focus on animal studies permits insight into the histological assessment of repair morphology. In summary, we aim to investigate the degree of integrative repair produced by BMSCs while also paying due attention to the quality of the repair tissue formed.

## 2. Results

### 2.1. Study Characteristics

Details of the 12 included studies, including study designs, subject demographics, lesion characteristics, and interventions are displayed in Table 1 [22,23,24,25,26,27,28,29,30,31,32,33]. All of the included studies were RCTs investigating the treatment of surgically created defects of femoral cartilage. Most involved the patellar groove, while two focused on defects of the medial femoral condyle [22,23], and one on the intercondylar fossa [29]. Defect sizes varied from 3.2 to 6 mm in diameter and 1.5 to 5 mm in depth in studies using rabbits. One study with horses as subjects involved treatment of defects 15 mm in diameter [32]. Seven studies used BMSCs harvested from the tibial bone marrow [22,24,26,27,28,29,31], four from the ilium [23,25,32,33], and one from the femoral metaphysis [30]. One study used periosteum-, synovium-, adipose-, and muscle-derived MSCs as comparators [30]. The number of cells per implant varied from an order of magnitude of 10^5^ [29] to 10^11^ [26,27,28] in the studies which recorded this parameter. Protein scaffolds or gels were commonly based on collagen [22,24], hyaluronic acid [23,25], glycolic acid [26,27,28], or caprolactone polymers [31,33]. Exceptions included one study using platelet-enhanced fibrin [32] and another using demineralized bone as a scaffold [30]. Most studies implanted undifferentiated BMSCs into the chondral defects, while a minority investigated BMSCs differentiated into chondrocytes in vitro prior to implantation [27,28,33]. The timing of the final sacrifice of the animal subjects was at least three months for most studies and ranged between two [24] and twelve months [32].

### 2.2. Histological Scores

The results of the quantitative histological assessments are displayed in Table 2. Integration, total histology scores, and indicators of repair cartilage resemblance to hyaline cartilage (cell morphology and matrix staining) are reported. Six studies used the Wakitani scale or a modification [22,24,25,26,27,28], three used the (modified) O’Driscoll system [23,30,33], and another three used an unnamed scale assessing similar constructs [29,31,32].

#### 2.2.1. Integration of Repair Tissue

Studies are divided into two groups based on the comparable reporting of integration outcomes. While eleven studies assessed integration using the aforementioned three-point scale, four recorded better outcomes as higher scores [23,29,32,33], and seven reported lower scores as an indication of better outcomes [22,24,25,26,27,28,30]. A single study was not comparable to others [31]. Meta-analysis of the former group (where 2/2 = both edges integrated and 0/2 = no integration) demonstrated a pooled raw mean (MRAW) integration score of 1.50 (95% CI (1.02, 1.99)) at the point of final sacrifice (Figure 1). The forest plot (Figure 2) derived from the meta-analysis of the second group (0/2 = both edges integrated, 2/2 = no integration) showed that MRAW was equal to 0.63 (95% CI (0.39, 0.87)).

Of the studies providing serial histology values over time [22,24,25,26,27,28,29,31], nearly all demonstrated that integration improved throughout the study period. Only two cohorts demonstrated static integration values between first and final sacrifice [25,29]. None showed a decline in integration quality.

Except for one [25], all studies showed that BMSCs generated superior integration results in comparison to cell-free controls; five of these showed a statistically significant difference [26,27,28,30,33]. Other comparators included MSCs from different sources [22,30]. Wakitani et al. used collagen gel embedded with MSCs for defect repair [22]. There was no difference in the quality of implant integration when comparing periosteum-derived MSCs (PMSCs) and BMSCs at 24 weeks. In contrast, Li et al. demonstrated superior integration of implant and host cartilage when using BMSCs [30]. After embedding all MSC types in demineralized bone, they demonstrated better outcomes in comparison to periosteum-, adipose-, synovium-, and muscle-derived MSCs. This difference was statistically significant (*p* < 0.05).

Studies also differed regarding the composition of scaffolds or gels. Fan et al. directly compared the use of scaffolds composed of poly-(lactic-co-glycolic acid) (PLGA) with hybrid PLGA–gelatin/chondroitin/hyaluronate (PLGA–GCH) scaffolds [26]. Integration did differ between the groups at 24 weeks, the time of final sacrifice. The PLGA-GCH group showed better quality integration in comparison to the PLGA group. While the difference was not statistically significant, the former showed significantly better integration in comparison to the empty control cohort. A sufficient number of studies assessed integration comparably and were, therefore, amenable to subgroup analysis on the basis of scaffold type (Figure 3). Analysis demonstrated a statistically significant difference in favor of PLGA-based scaffolds (*p* = 0.001). While it was not valid to include Zhou et al.’s study in subgroup analysis, their study investigating polyglycolic-acid-hydroxyapatite (PLGA-HA) scaffolds demonstrated better integration in comparison to the cell-free scaffold and empty control groups. However, these differences were not statistically significant.

Limited study numbers and variability in the recording of results precluded the completion of further subgroup analyses on the basis of, for example, scaffold or gel composition, cell density, or timing of sacrifice. Furthermore, the impacts of varying such factors were rarely investigated directly in the selected studies. Three studies did directly compare in vitro and in vivo differentiation methods [27,28,33]. Wu et al.’s study used BMSCs embedded in polycaprolactone polymeric films [33]. Both the mixed phenotype (undifferentiated and pre-differentiated) and bilayered pre-differentiated BMSC implants exceeded the integration quality observed in the undifferentiated group. In the case of the bilayered differentiated group, this difference was statistically significant (*p* < 0.05). The other two studies also showed statistically significant differences in the integration quality achieved by in vitro and in vivo differentiated BMSCs [27,28]. However, this difference was in favor of the in vivo group.

#### 2.2.2. Other Histology Scores

Scores pertaining to repair tissue morphology and staining characteristics are represented in Table 1, as are the total histological scores. Morphology and matrix staining both showed similar trends to integration, with the performance of BMSC interventions exceeding that of cell-free scaffolds and empty controls across all studies. This difference was statistically significant for four of the ten studies making this comparison [23,28,30,31]. Furthermore, increased post-operative time was associated with an improvement in these scores.

While total histological scores were similar between two BMSC cohorts and their respective cell-free comparators [26,28], the remaining majority of BMSC interventions consistently showed superior total scores in comparison to cell-free interventions or controls. This difference was statistically significant for six of the nine studies making this comparison [23,24,25,26,28,29,30,31]. Again, all studies showed a sustained and improved total score as the time between the intervention and sacrifice increased.

Li et al. compared the performance of MSCs from various sources loaded on a demineralized bone scaffold [30]. They demonstrated that the performance of BMSCs with regard to repair tissue cell morphology and staining properties exceeded that of periosteum-, adipose-, synovium-, and muscle-derived MSCs to a statistically significant degree. The same is true for the total scores. In contrast, Wakitani et al.’s comparison between BMSCs and PMSCs did not show a difference in any of these categories [22].

While no study directly compared wholly different polymer scaffolds, Fan et al. showed that the hybridization of PLGA and a gelatin/chondroitin/hyaluronate component led to improved outcomes [26]. On assessment of cell morphology and matrix staining, the PLGA-GCH/MSC group showed significantly better results compared with the PLGA/MSC group at 24 weeks (*p* < 0.05). Total scores also differed significantly at 12 and 24 weeks (*p* < 0.05).

Three studies directly compared undifferentiated and pre-differentiated BMSCs [26,28,33], with inconsistent results. Two of these showed better morphology, staining, and total scores for in vivo differentiated BMSCs in comparison to in vitro differentiated groups when using poly-lactic-co-glycolic acid (PLGA) scaffolds [26,28]. Twice the difference in total scores was statistically significant, and on one occasion, morphology and staining were significantly better in vivo compared to in vitro differentiation. Conversely, using polycaprolactone scaffolds, Wu et al. demonstrated that pre-differentiated BMSCs yielded superior performances in comparison to cells undifferentiated at the time of implantation [33]. Both the mixed phenotype (undifferentiated and pre-differentiated) and bilayered pre-differentiated BMSC implants had higher total scores than the undifferentiated group. Morphology and staining were not assessed.

### 2.3. Risk of Bias Assessment

Overall, there were only low or some concerns with regard to bias in the included studies (Figure 4). The lack of an explicit statement regarding whether study subjects were randomized into each treatment arm mostly contributed to concerns. In addition, experimenters carrying out the interventions unavoidably were aware of whether animals received an implant or the cartilage defect remained empty. Otherwise, almost all studies included outcomes for all or nearly all participants, and most had blinded investigators for histological scoring.

## 3. Discussion

This review aimed to investigate the integration of native and implant cartilage, utilizing the results of animal studies using BMSC-seeded implants for the repair of focal cartilage defects of the knee. The quality of this integration is an important factor in determining the effectiveness of implants in repairing chondral defects [34]. However, conclusions regarding cartilage–cartilage integration in human knees following the use of MSC-seeded implants have been limited by a lack of clinical studies and small sample sizes [35]. The small number of human studies investigating MSC-based therapies have demonstrated promising results with regard to both integration and clinically observable outcomes [36]. None of the studies included in this review directly compared MSCs to ACI. However, MSC implant therapies have been shown to outperform the more well-established ACI technique in humans, in terms of both functional parameters and the appearance of repair tissue on MRI [37]. The potential benefits of MSC therapies warrant further investigation. The relative abundance of animal studies necessitated that these became the focus of this review.

Twelve studies were included, reporting quantitative results of histological outcomes following BMSC implantation. Overall, BMSC-seeded implants achieved reasonably good quality integration. Pooled integration scores obtained through meta-analyses equated to more than one of the two sides of the implant under investigation that achieved integration with native cartilage. Furthermore, BMSC implants had better integration scores compared to controls, cell-free implants, and MSCs from other sources. Serial measurements also demonstrated that integration quality improved over time. Interestingly, the use of a PLGA scaffold was found to achieve better quality integration in comparison to other implants to a statistically significant degree.

Despite these promising results, heterogeneity in interventions has made it difficult to draw conclusions on the best implant engineering techniques, a problem mirrored in the literature [38]. Furthermore, long-term follow-up to determine the functional benefit of MSC implants need to be addressed if these animal studies are ultimately to inform human studies and clinical practice.

### 3.1. Study Heterogeneity

#### 3.1.1. MSC Source

The heterogeneity of MSC phenotypes is a well-documented issue in studies investigating their use in cartilage regeneration [37]. The success of implant integration relies on a multitude of biomechanical and joint environmental factors, many dependent on the tissue-engineering processes involved in cultivating, differentiating, culturing, and seeding MSCs onto the implants [34].

To eliminate some of this variety, this review focused solely on bone marrow-derived MSCs. Owing to their relatively well-developed and common use in the repair of animal cartilage [39] and in human trials [40], BMSCs are also likely to be the most translatable cell type for human studies. Nonetheless, the 12 studies did include three different bone sources of BMSCs. The tibial bone marrow was the most common (7/12 studies), followed by the iliac crest (4/12 studies), and one study used the femur. Therefore, heterogeneity is not eliminated, as BMSCs cultivated from different areas possess different properties which may predispose them for better cartilage regeneration [38]. For instance, Hermann et al. showed that iliac crest chondrocyte cells were superior to femoral head cells in chondrogenic potential [41].

Comparisons between different MSC sources in our included studies could not be made as there was insufficient comparably reported data to perform subgroup analyses. Further investigation with larger sample sizes comparing MSCs from different sites will be essential for the future selection of the most appropriate source for cartilage repair.

The selection of MSCs through the expression of cell surface markers may play a future role in reducing heterogeneity even further. Studies investigating this have shown that a variety of different markers are expressed even amongst MSCs cultivated from the same area of the body [38]. The studies in this review did not quality control MSCs by cell surface marker expression, but this has been investigated in human clinical trials. Akgun et al. used flow cytometry to select MSCs for cluster of differentiation 105 (CD105), CD73, and CD90 and lack surface expression of CD45, CD34, CD14 (or CD11 b), CD79 a (or CD19), and Human Leukocyte Antigen–DR (HLA-DR) [37], following the standards stated in the Mesenchymal and Tissue Stem Cell Committee of the International Society for Cellular Therapy. Additional novel surface markers are now being used to characterize MSCs, with certain markers showing an association with the cell’s ability to regenerate cartilage [38]. This means obtaining a molecular profile will be especially pertinent if future studies are to determine which profiles are associated with the best integration.

In addition to the MSC source, several papers investigated if factors such as differentiation status [27,28,33] or transfection of genes [24] would affect integration. Again, there was an insufficient number of papers to perform subgroup analyses. Future follow-up studies with larger sample sizes are be needed to establish how important these factors are to cartilage integration.

#### 3.1.2. Implant Composition

Another source of heterogeneity was in the composition of implants in which MSCs were embedded. These were largely composed of either protein or organic acid polymers. In our preceding systematic review, mainly focused on the use of ACI in humans, we found through subgroup analysis that collagen scaffolds were associated with significantly better outcomes across multiple clinical scoring systems when compared to other scaffold types [4]. Integration, assessed using magnetic resonance imaging (MRI), was also superior, although this was not statistically significant (*p* = 0.06). Given these findings, we investigated whether scaffold composition had any bearing on integration outcomes when using BMSCs in animal studies. Subgroup analysis showed that the use of PLGA scaffolds was associated with a statistically significant difference in integration quality when compared to other scaffold types.

The findings of subgroup analyses in this and our previous paper demonstrate that the choice of polymer used for implant composition has an important influence on the outcome of regenerative cell therapies. Different scaffolds possess differing mechanical properties and have variable effects on cell expansion, differentiation, and retention within implants [42,43]. Graft survival may also be influenced by immunoreactivity to synthetic materials, like in the case of PLGA, which needs to be balanced against the potentially inconsistent quality of more benign natural materials such as collagen. Combinations of biomaterials also exist in numerous possible permutations. For example, combining PLGA with fibrin and HA may be more effective than PLGA alone in promoting chondrogenesis [42].

Among the studies included in our review, the best integration was achieved with the PLGA-gelatin/chondroitin/hyaluronate (PLGA- GCH) hybrid scaffold combined with gelatin microspheres impregnated with transforming growth factor-β1 [36]. Therefore, studying the performance of different scaffold types is as significant as the choice of cells used. The large-scale assessment of various combinations of cell type, core scaffold composition, and added components, ideally in human patients, represents a priority for future research in this domain.

In summary, heterogeneity in techniques has made it difficult to select the best processes for maximizing integration, a problem which must be resolved by future human trials.

### 3.2. Translation to Clinical Practice

This review focused on animal studies due to the relative lack of human studies investigating MSC implant integration [4]. Not only are such results more widely reported by animal studies, but these papers also offer insight into histological appearances of cartilage repair, which is rarely the case in human studies. There are obvious barriers to the translation of these animal studies into clinical outcomes. Differences in cartilage geometrical and mechanical properties between different species is a limitation when extrapolating the findings.

An additional value of human studies is afforded by being able to assess functional outcomes, which are possibly the highest priority for patients themselves. Comments on functional outcomes were limited in the included studies. In two studies, it was observed that the animals did not display an abnormal gait following the procedure [22,29]. This was despite immediate post-operative weight bearing, an important difference compared to human studies, which commonly involve external fixation and gradual weight-bearing regimens [22,44]. Bornes et al. [45] summarize studies reporting level IV evidence regarding the efficacy of BMSCs in humans. All of these demonstrated either a return to baseline levels of activity or improvements in validated clinical scores. Furthermore, a phase I randomized controlled trial comparing BMSCs embedded in atelocollagen scaffolds to microfracture demonstrated improved International Knee Documentation Committee (IKDC) scores for both techniques [46]. While the BMSC group showed a sustained improvement at the two-year follow-up, the IKDC scores of those treated with microfracture demonstrated a subsequent decline before one year, which continued until the final follow-up. Gobbi et al. [47] compared the efficacy of bone marrow aspirate concentrate (BMAC), containing BMSCs, and MACI, both using a hyaluronan scaffold. Improving clinical scores were seen from baseline to two years post-operation. Participants treated with BMAC showed a continued improvement between two years and final follow-up, while the MACI group showed declining function. Significant improvements in functional scores after the implantation of BMAC were seen in a further two studies conducted by the same group [48,49].

These studies exemplify the importance of long-term follow-up data for demonstrating the safety and long-term efficacy of interventions for cartilage repair, another feature lacking from animal studies. Within the twelve included studies, only one had a follow-up duration longer than one year [32]. The importance of these data is signified by our previous review, in which the longevity of implants was called into question [4]. Declining trends in imaging scores were sometimes only appreciable several years after treatment with MACI.

Data from the MRI assessment of cartilage repair following implantation of BMSCs in humans are limited in availability but have shown encouraging results. Gobbi and colleagues [47,48,49] have consistently demonstrated high-quality repair following at least three years of follow up after the implantation of BMAC on polymer scaffolds. Imaging from a large majority of their patients has shown complete or near complete defect filling and complete integration with adjacent host cartilage. While the authors did not statistically correlate clinical scores with a quantitative assessment of the repair, using the Magnetic Resonance Observation of Cartilage Repair Tissue (MOCART) score [36], for example, they commented that good quality repair, as seen on MRI, was consistent with promising clinical outcomes.

## 4. Materials and Methods

### 4.1. Search Algorithm

This review was conducted in accordance with the Preferred Reporting Items for Systematic Reviews and Meta-Analysis (PRISMA guidelines) [50]. A comprehensive literature search was conducted from conception to October 2022 using the following databases: (1) PubMed, (2) Embase, (3) MEDLINE, (4) Web of Science, and (5) CINAHL. The detailed search strategy can be found in Appendix A. This review was registered in the International Prospective Register of Systematic Reviews PROSPERO (CRD42022343052). Studies were uploaded onto the Rayyan website [51], where titles and abstracts were independently screened by EL and IEE before a subsequent full-text screen. A third (VL) and fourth reviewer (WK) were consulted for unresolvable disagreements.

### 4.2. Inclusion and Exclusion Criteria

Using the Population, Intervention, Comparison, Outcome, Study type (PICOS) model [52] as a guide, we formulated our inclusion and exclusion criteria for study selection (Table 3).

### 4.3. Search Results

The structured search, using five databases, yielded 948 papers in total (Figure 5). After removal of duplicates, 829 articles remained, of which 773 were excluded following title and abstract screening. Of the remaining 56 studies which underwent full-text screening, 12 were eligible for inclusion.

### 4.4. Data Extraction

Extraction of the data was independently performed by EL and IEE, using third and fourth reviewers (WK and VL) to resolve disagreements. A standardized table created in an Excel spreadsheet was populated with extracted data including:Study characteristics and demographics including study design, animal type, cohort size, cartilage defect location, defect size, and time of sacrifice.Type of intervention including source of BMSC and scaffold composition.The cluster of differentiation (CD) molecule profile of cells.Primary outcome measures regarding integration of the implant and native cartilage, assessed by histological scoring systems.Secondary outcomes including total histological scores, data regarding the cartilage morphology, and qualitative descriptions of macro- and microscopic cartilage characteristics.

### 4.5. Data Analysis

Histological scores were recorded in a format defined by the grading scale used in each paper. These were namely original or modified versions of the Wakitani [22] and O’Driscoll [23] grading scales. Occasionally, papers use unnamed scales. These largely assessed similar constructs, including cartilage morphology, staining characteristics, underlying subchondral bone quality, surface regularity, defect fill, integration, and degenerative changes.

Integration quality was usually assessed on a three-point scale: (a) both edges integrated, (b) one edge integrated, or (c) neither edge integrated. Points given (0, 1 or 2) depended on whether ascending or descending scores were assigned to a better or worse outcome. When comparable scoring systems were used, outcomes of integration of regenerated and host cartilage were pooled in meta-analyses.

Total histological, cartilage morphology, and staining scores were considered pertinent to our understanding of the quality of cartilage repair and were, therefore, extracted. Due to variations in the recording of these assessments, they were not pooled in meta-analysis but analyzed qualitatively for trends.

Meta-analyses were carried out using RStudio version 4.0.5. for continuous data. The estimator reported by Hozo et al. was used where the standard deviation was not provided in the manuscript [53]. Higgins and Thompson’s I2 statistic and Cochran’s Q test were used as measures of heterogeneity [54,55]. Prediction intervals were also included to provide a range into which future studies’ effect size can be expected to fall into.

### 4.6. Assessing Risk of Bias

Risk of bias assessments were carried out independently by EL and IE, and VL was consulted for unresolvable disagreements. The Cochrane RoB 2.0 tool was used to assess the randomized trials according to its five domains [56]: (1) bias from the randomization process; (2) bias due to deviations from the intended interventions; (3) missing outcome data; (4) bias in measurement of the outcome; and (5) bias in selection of the reported result. These domains were each assessed as having low risk, some concerns, or high risk of bias, and an overall risk was determined. Results of the assessments were presented using the robvis package in RStudio [57].

## 5. Conclusions

This review collated results of integration outcomes following the repair of focal cartilage defects of animal knees using BMSC-seeded implants. BMSC-seeded implants achieved good quality integration with the native cartilage. Studying integration in animals allows the establishment of a baseline by which investigations of BMSC-based treatments in humans can be informed. We also hope to reiterate future areas of focus for the optimization of this promising therapy. Future studies could focus on the standardization of techniques to reduce heterogeneity, namely cell source, implant composition, and molecular characterization of MSCs, to achieve an understanding of the best combination of these various factors.

## Figures and Tables

**Figure 1 ijms-24-03227-f001:**
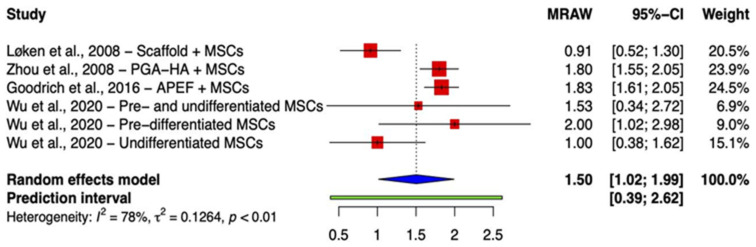
Forest plot on the mean histological integration score after receiving BMSC implant therapy, where 2/2 points = both edges integrated, 1/2 = one edge integrated, and 0/2 = no integration. (Abbreviations: MSC, mesenchymal stem cell; PGA-HA, polyglycolic acid-hydroxyapatite; APEF, autologous platelet-enhanced fibrin; CI, Confidence Intervals) [23,29,32,33].

**Figure 2 ijms-24-03227-f002:**
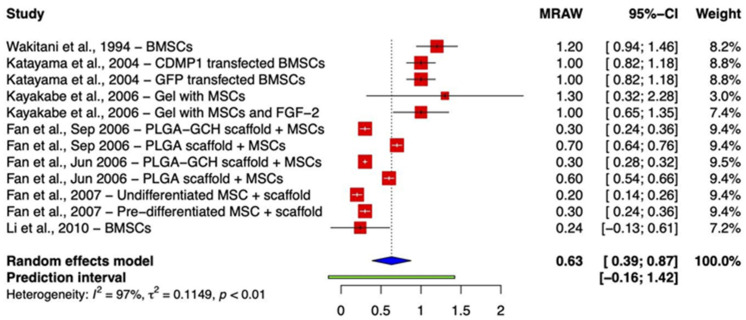
Forest plot on the mean histological integration score after receiving BMSC implant therapy, where 0/2 points = both edges integrated, 1/2 = one edge integrated, and 2/2 = no integration. (Abbreviations: BMSC, bone marrow-derived mesenchymal stem cell; CDMP1, cartilage-derived morphogenetic protein 1; GFP, green fluorescent protein; FGF-2, fibroblast growth factor-2; PLGA, poly-(lactic-co-glycolic acid); GCH, gelatin/chondroitin/hyaluronate; CI, Confidence Intervals) [22,24,25,26,27,28,30].

**Figure 3 ijms-24-03227-f003:**
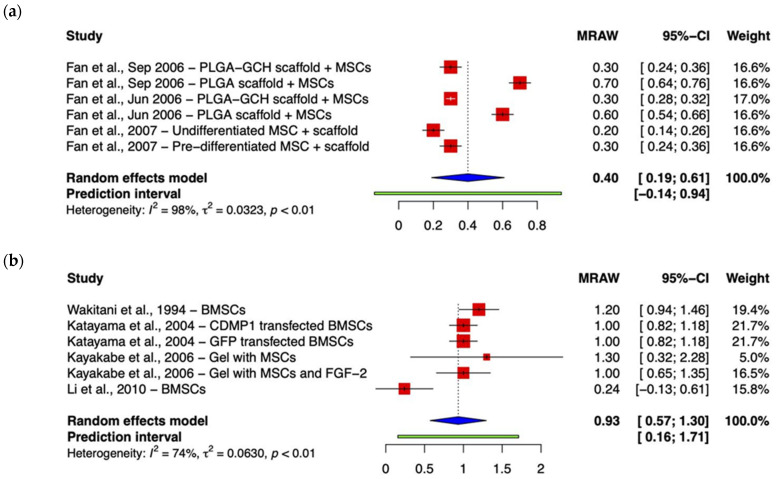
Forest plots representing subgroup meta-analyses of mean integration scores depending on whether scaffolds were composed of (**a**) PLGA or (**b**) other polymers. Lower scores equate to better outcomes, where 0/2 points = both edges integrated, 1/2 = one edge integrated, and 2/2 = no integration. (Abbreviations: BMSC, bone marrow-derived mesenchymal stem cell; CDMP1, cartilage-derived morphogenetic protein 1; GFP, green fluorescent protein; FGF-2, fibroblast growth factor-2; PLGA, poly-(lactic-co-glycolic acid); GCH, gelatin/chondroitin/hyaluronate; CI, Confidence Intervals) [22,24,25,26,27,28,30].

**Figure 4 ijms-24-03227-f004:**
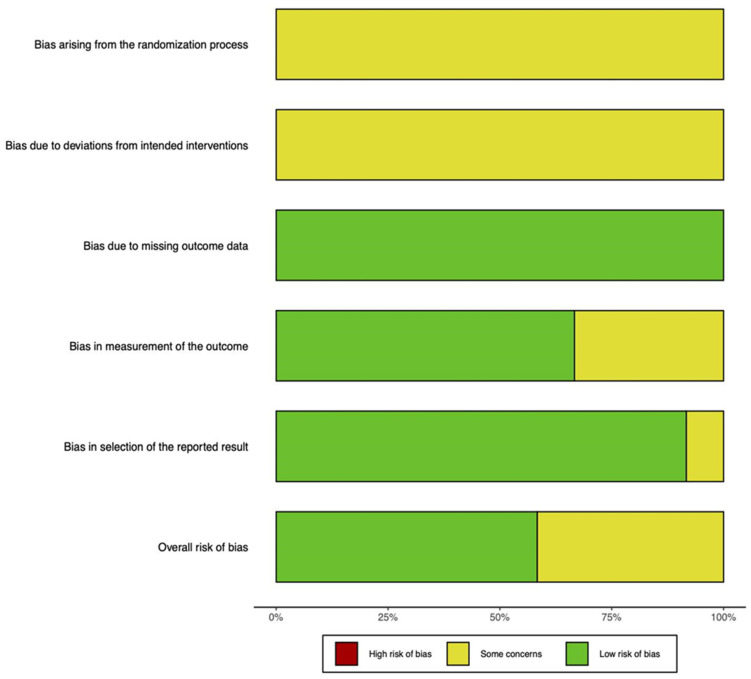
Summary graph showing the overall risk of bias analysis using the RoB 2.0 tool in randomized studies.

**Figure 5 ijms-24-03227-f005:**
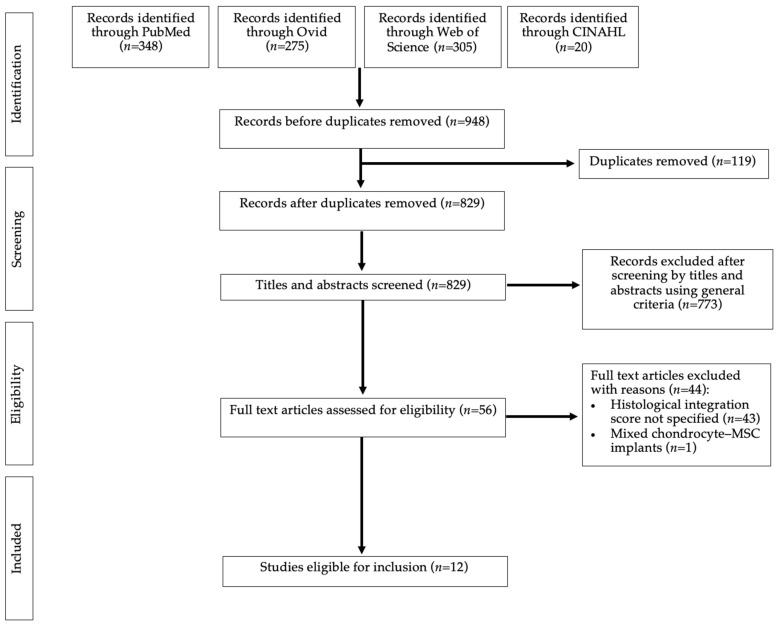
A PRISMA flow-chart representing article screening and selection.

**Table 1 ijms-24-03227-t001:** Demographics and Study Design.

Author	Animal Model	Cohort Size	Defect Location and Size	Scaffold Composition	MSC Source	Cell Density	Differentiation	Interventions	Timing of Sacrifice
Wakitani et al., 1994 [22]	Rabbits	68	Medial femoral condyle, 6 mm × 3 mm × 3 mm	Type I collagen gel	Tibial bone marrow or periosteum	5 × 10^6^ cells per mL	in vivo	Collagen gel + bone marrow-derived MSCs, collagen gel + periosteum-derived MSCs, cell-fee collagen gel, empty defect	2, 4, 12, and 24 weeks
Katayama et al., 2004 [24]	Rabbits	46	Patellar groove, 4 mm diameter × 4 mm depth	Type I collagen gel	Tibial bone marrow	1 × 10^6^ cells per 200 µL of gel	in vivo	Green fluorescent protein gene-transfected bone marrow-derived mesenchymal cells, cartilage-derived morphogenetic protein 1 gene-transfected bone marrow-derived mesenchymal cells, empty control	2, 4, and 8 weeks
Kayakabe et al., 2006 [25]	Rabbits	54	Patellar groove, 5 mm diameter × 2–3 mm depth	Hyaluronic acid gel sponge or atelocollagen gel	Ilium bone marrow	1 × 10^6^ cells per 100 µL of Dulbecco’s Modified Eagle Medium	in vivo	Cell-free hyaluronic gel sponge, sponge with autologous MSC grown without fibroblast growth factor-2 (FGF-2), sponge with autologous MSC grown with FGF-2, atelocollagen + MSC implant, empty control	4 and 12 weeks
Fan et al., 2006 [26]	Rabbits	45 (90 knees)	Patellar groove, 4 mm diameter, 3 mm depth	Poly-lactic-co-glycolic acid (PLGA) scaffold or hybrid PLGA-gelatin/chondroitin/hyaluronate scaffold (PLGA-GCH)	Tibial bone marrow	1 × 10^11^ cells per scaffold	in vitro	PLGA scaffold + MSCs, PLGA-GCH scaffold + MSCs, empty control	6, 12, and 24 weeks
Fan et al., 2006 [27]	Rabbits	30	Patellar groove, 4 mm diameter, 3 mm depth	Porous PLGA-GCH	Tibial bone marrow	1 × 10^11^ cells per scaffold	in vitro and in vivo	Porous PLGA-GCH scaffold + undifferentiated MSCs, PLGA scaffold + pre-differentiated MSCs, empty control	6, 12, and 24 weeks
Fan et al., 2007 [28]	Rabbits	30	Patellar groove, 4 mm diameter, 3 mm depth	PLGA scaffold with transforming growth factor-β1-impregnated microspheres (PLGA-GCH/MS-TGF)	Tibial bone marrow	1 × 10^11^ cells per scaffold	in vitro and in vivo	Undifferentiated MSC + PLGA-GCH/MS-TGF scaffold, pre-differentiated MSC + PLGA-GCH/MS-TGF scaffold, empty control	6, 12, and 24 weeks
Løken et al., 2008 [23]	Rabbits	11 (22 knees)	Medial femoral condyle, 4 mm × 1.5 mm	HYAFF-11^®^ (Fidia Advanced Biopolymers, Abano Terme, Italy)	Iliac crest bone marrow	10^7^ per cm^2^	in vivo	Scaffold + MSC, cell-free scaffold	24 weeks
Zhou et al., 2008 [29]	Rabbits	30	Femoral intercondylar fossa, 3.2 mm diameter × 3 mm depth	Polyglycolic-acid-hydroxyapatite (PLGA-HA)	Tibial bone marrow	4 × 10^5^ per scaffold	in vivo	PLGA-HA + MSCs, cell-free PLGA-HA, empty control	16 and 32 weeks
Li et al., 2010 [30]	Rabbits	54 (108 knees)	Femoral trochlear, 4 mm diameter × 3 mm depth	Demineralized bone	Femoral metaphysis, tibial periosteum, knee synovium, adipose perinephric fat, anterior tibial muscle	NA	in vivo	Bone marrow-derived MSCs, periosteum-derived MSCs, synovium-derived MSCs, adipose-derived MSCs, muscle-derived MSCs, empty control	4, 8, and 12 weeks
Xie et al., 2010 [31]	Rabbits	18 (36 knees)	Medial femoral condyle, 4.5 mm diameter × 5 mm depth	poly(L-lactide-co-ε-caprolactone) (PLCL)	Tibial bone marrow	NA	in vivo	PLCL + MSCs, cell-free PLCL, empty control	3 and 6 months
Goodrich et al., 2016 [32]	Horses	12 (24 joints)	Stifle Joint, 15 mm diameter, full thickness	Autologous platelet-enhanced fibrin (AFEP)	Ilium bone marrow	10^6^ per mL	in vivo	APEP + MSCs, cell-free APEP	3 and 12 months
Wu et al., 2020 [33]	Rabbits	24	Trochlear groove, 4.5 mm diameter × 2 mm depth	Nanoimprinted polymeric films with polycaprolactone	Iliac crest	NA	in vitro and in vivo	Mixed pre- and undifferentiated MSCs, bilayered pre-differentiated MSCs, undifferentiated MSCs, empty control	7 months

**Table 2 ijms-24-03227-t002:** Histology scores.

Author	Scoring	Intervention	Number of Subjects	Total Score	Integration	Cell Morphology	Matrix Staining	*p*
Wakitani et al., 1994 [22]	Wakitani scale (lower is better): 0 = both edges integrated, 1 = one edge integrated, 2 = neither edge integrated	Collagen gel + bone marrow derived-MSCs	7 (2 weeks); 8 (4 weeks); 9 (12 weeks); 7 (24 weeks)	9.8 (2 weeks); 5.6 (4 weeks); 7.9 (12); 8.4 (24 weeks)	1.6 (2 weeks); 1.1 (4 weeks); 1.2 (24 weeks)	2.6 (2 weeks); 1.3 (4 weeks); 1.9 (12 weeks); 1.9 (24 weeks)	2.2 (2 weeks); 1.3 (4 weeks); 1.7 (12 weeks); 1.7 (24 weeks)	Not assessed
		Collagen gel + periosteum-derived MSCs	6 (2 weeks); 17 (4 weeks); 8 (12 weeks); 6 (24 weeks)	9.3 (2 weeks); 6.0 (4 weeks); 8.1 (12 weeks); 8.4 (24 weeks)	1.5 (2 weeks); 1.2 (4 weeks); 1.5 (12 weeks); 1.2 (24 weeks)	2.5 (2 weeks); 1.4 (4 weeks); 1.7 (12 weeks); 2.0 (24 weeks)	2.0 (2 weeks); 1.3 (4 weeks); 1.6 (12 weeks); 1.6 (24 weeks)	
		Cell-free collagen gel	11 (2 weeks); 19 (4 weeks); 12 (12 weeks); 7 (24 weeks)	11.6 (2 weeks); 6.9 (4 weeks); 8.7 (12 weeks); 8.9 (24 weeks)	1.9 (2 weeks); 0.9 (4 weeks); 1.6 (12 weeks); 1.2 (24 weeks)	2.9 (2 weeks); 1.7 (4 weeks); 1.8 (12 weeks); 2.2 (24 weeks)	2.2 (2 weeks); 1.6 (4 weeks); 1.7 (12 weeks); 2.0 (24 weeks)	
		Empty control	5 (2 weeks); 7 (4 weeks); 4 (12 weeks); 3 (24 weeks)	11.8 (2 weeks); 9.0 (4 weeks); 9.2 (12 weeks); 9.5 (24 weeks)	1.9 (2 weeks); 1.8 (4 weeks); 1.3 (12 weeks); 1.4 (24 weeks)	3.2 (2 weeks); 1.9 (4 weeks); 2.2 (12 weeks); 2.4 (24 weeks)	2.2 (2 weeks); 1.8 (4 weeks); 1.8 (12 weeks); 1.9 (24 weeks)	
Katayama et al., 2004 [24]	Modified Wakitani scale (lower is better): 0 = both edges integrated, 1 = integrated at one edge, 2 = no integration	Cartilage-derived morphogenetic protein 1 gene-transfected bone marrow-derived mesenchymal cells	10 (2 weeks); 10 (4 weeks); 10 (8 weeks)	11.5 (2 weeks); 7.4 (4 weeks); 7.8 (8 weeks)	1.2 (2 weeks); 0.7 (4 weeks); 1.0 (8 weeks)	Cell morphology and staining composite score: 6.2 (2 weeks); 4.4 (4 weeks); 4.6 (8 weeks)		Total: *p* < 0.05 between CDMP group and empty control at 2 weeks; *p* < 0.05 between CDMP group and the other groups at 8 weeks
		Green fluorescent protein gene-transfected bone marrow-derived mesenchymal cells	10 (2 weeks); 10 (4 weeks); 10 (8 weeks)	13.4 (2 weeks); 10.7 (4 weeks); 12.7 (8 weeks)	1.4 (2 weeks); 0.9 (4 weeks); 1.0 (8 weeks)	7.0 (2 weeks); 6.2 (4 weeks); 6.8 (8 weeks)		
		Empty control	2 (2 weeks); 7 (4 weeks); 7 (8 weeks)	18.5 (2 weeks); 12.7 (4 weeks); 13.6 (8 weeks)	2.0 (2 weeks); 1.6 (4 weeks); 1.0 (8 weeks)	8.0 (2 weeks); 6.6 (4 weeks); 7.4 (8 weeks)		
Kayakabe et al., 2006 [25]	Modified Wakitani scale (lower is better): 0 = both edges integrated, 1 = integrated at one edge, 2 = no integration	Cell-free hyaluronic gel sponge alone	6 (4 weeks); 3 (12 weeks)	9.2 ± 2.5 (4 weeks); 6.3 ± 1.2 (12 weeks)	1.3 ± 0.5 (4 weeks); 1.7 ± 0.6 (12 weeks)	2.0 ± 0.0 (4 weeks); 1.3 ± 0.6 (12 weeks)	1.3 ± 0.6 (4 weeks); 1.0 ± 0.0 (12 weeks)	Total: *p* < 0.05 between HS/MSC/ FGF and empty control at 12 weeks
		Hyaluronic acid gel sponge loaded with autologous MSC grown without FGF-2	3 (4 weeks); 4 (12 weeks)	7.7 ± 0.6 (4 weeks); 5.8 ± 1.5 (12 weeks)	2.0 ± 0.0 (4 weeks); 1.3 ± 1.0 (12 weeks)	2.3 ± 0.8 (4 weeks); 1.5 ± 0.6 (12 weeks)	2.0 ± 0.6 (4 weeks); 1.3 ± 0.5 (12 weeks)	
		Hyaluronic gel sponge loaded with autologous MSC grown with FGF-2	9 (4 weeks); 8 (12 weeks)	8.7 ± 3.1 (4 weeks); 4.0 ± 1.4 (12 weeks)	1.0 ± 1.0 (4 weeks); 1.0 ± 0.5 (12 weeks)	2.7 ± 0.6 (4 weeks); 2.8 ± 0.5 (12 weeks)	2.7 ± 0.6 (4 weeks); 0.5 ± 0.8 (12 weeks)	
		Atelocollagen loaded with autologous MSC grown with FGF-2	NA	7.8 ± 3.0 (4 weeks); 5.0 ± 2.0 (12 weeks)	NA	NA	NA	
		Empty control	3 (4 weeks); 4 (12 weeks)	9.7 ± 2.5 (4 weeks); 8.5 ± 1.3 (12 weeks)	1.3 ± 0.6 (4 weeks); 1.0 ± 0.0 (12 weeks)	2.4 ± 1.3 (4 weeks); 2.8 ± 0.5 (12 weeks)	2.2 ± 1.0 (4 weeks); 2.0 ± 0.0 (12 weeks)	
Fan et al., 2006 [26]	Modified Wakitani scale (lower is better): 0 = both edges integrated, 1 = integrated at one edge, 2 = no integration	PLGA-GCH scaffold + MSCs	10 (6 weeks); 10 (12 weeks); 10 (24 weeks)	7.7 ± 0.2 (6 weeks); 4.7 ± 0.2 (12 weeks); 3.7 ± 0.1 (24 weeks)	1.0 ± 0.2 (6 weeks); 0.4 ± 0.1 (12 weeks); 0.3 ± 0.1 (24 weeks)	4.5 ± 0.4 (6 weeks); 2.9 ± 0.3 (12 weeks); 2.2 ± 0.1 (24 weeks)		Integration: *p* < 0.05 for PLGA-GCH/MSC group compared with control at 12 and 24 weeks Total: *p* < 0.05 for PLGA-GCH/MSC group compared with control at 6, 12 and 24 weeks; *p* < 0.05 for PLGA-GCH/MSC group compared with PLGA/ MSC group at 12 and 24 weeks Cell morphology and matrix staining: *p* < 0.05 for PLGA-GCH/MSC group compared with control at 12 and 24 weeks; *p* < 0.05 for PLGA-GCH/MSC group compared with PLGA/ MSC group at 24 weeks
		PLGA scaffold + MSCs	10 (6 weeks); 10 (12 weeks); 10 (24 weeks)	8.3 ± 0.4 (6 weeks); 7.5 ± 0.3 (12 weeks); 8.2 ± 0.3 (24 weeks)	1.0 ± 0.2 (6 weeks); 0.8 ± 0.1 (12 weeks); 0.7 ± 0.1 (24 weeks)	4.9 ± 0.6 (6 weeks); 4.2 ± 0.3 (12 weeks); 4.7 ± 0.3 (24 weeks)		
		Empty control	10 (6 weeks); 10 (12 weeks); 10 (24 weeks)	17.3 ± 0.4 (6 weeks); 17.5 ± 0.3(12 weeks); 17.1 ± 0.5 (24 weeks)	2.2 ± 0.3 (6 weeks); 1.9 ± 0.1 (12 weeks); 1.6 ± 0.1 (24 weeks)	6.3 ± 0.8 (6 weeks); 7.5 ± 0.3(12 weeks); 7.6 ± 0.6 (24 weeks)		
Fan et al., 2006 [27]	Modified Wakitani scale (lower is better): 0 = both edges integrated, 1 = integrated at one edge, 2 = no integration	Porous PLGA-GCH scaffold + undifferentiated MSCs	10 (6 weeks); 10 (12 weeks); 10 (24 weeks)	7.9 ± 0.4 (6 weeks); 4.7 (12 weeks); 3.7 ± 0.2 (24 weeks)	1.0 ± 0.2 (6 weeks); 0.5 ± 0.1 (12 weeks); 0.3 ± 0.04 (24 weeks)	4.7 ± 0.5 (6 weeks); 3.0 ± 0.4 (12 weeks); 2.2 ± 0.1 (24 weeks)		Integration: *p* < 0.05 for in vivo repair group compared with control at 24 weeks Total: *p *< 0.05 compared with control at 6, 12 and 24 weeks and with the in vitro differentiated MSCs repair group at 12 and 24 weeks.
		PLGA scaffold + pre- differentiated MSCs	10 (6 weeks); 10 (12 weeks); 10 (24 weeks)	8.4 ± 0.4 (6 weeks); 7.4 ± 0.3 (12 weeks); 8.1 + 0.3 (24 weeks)	1.0 ± 0.3 (6 week); 0.8 ± 0.1 (12 weeks); 0.6 ± 0.1 (24 weeks)	5.0 ± 0.6 (6 weeks); 4.0 ± 0.7 (12 weeks); 4.7 ± 0.3 (24 weeks)		
		Empty control	10 (6 weeks); 10 (12 weeks); 10 (24 weeks)	17.2 ± 0.7 (6 weeks); 17.5 ± 0.4 (12 weeks); 16.9 ± 0.5 (24 weeks)	2.0 ± 0.3 (6 weeks); 1.8 ± 0.1 (12 weeks); 1.5 ± 0.1 (24 weeks)	6.5 ± 0.8 (6 weeks); 7.4 ± 0.3 (12 weeks); 7.6 ± 0.5 (24 weeks)		
Fan et al., 2007 [28]	Modified Wakitani scale (lower is better): 0 = both edges integrated, 1 = integrated at one edge, 2 = no integration	Undifferentiated MSC + PLGA-GCH/MS-TGF scaffold	10 (6 weeks); 10 (12 weeks); 10 (24 weeks)	7.2 + 0.2 (6 weeks); 4.2 ± 0.2 (12 weeks); 2.8 ± 0.1 (24 weeks)	0.9 ± 0.2 (6 weeks); 0.3 ± 0.1 (12 weeks); 0.2 ± 0.1 (24 weeks)	0.9 ± 0.2 (6 weeks); 0.3 ± 0.1 (12 weeks); 0.2 ± 0.1 (24 weeks)		Integration: *p* < 0.05 for in vivo differentiated group compared to control at 12 and 24 weeks. Morphology and Matrix Staining: *p* < 0.05 compared to control at 12 and 24 weeks, and compared to in vitro group at 24 weeks. Total: *p* < 0.05 compared to control at 6, 12, and 24 weeks, and compared to in vitro group at 12 and 24 weeks.
		Pre-differentiated MSC + PLGA-GCH/MS-TGF scaffold	10 (6 weeks); 10 (12 weeks); 10 (24 weeks)	8.0 ± 0.2 (6 weeks); 4.8 ± 0.2 (12 weeks); 4.0 ± 0.1 (24 weeks)	1.0 ± 0.2 (6 weeks); 0.5 ± 0.1 (12 weeks); 0.3 ± 0.1 (24 weeks)	1.0 ± 0.2 (6 weeks); 0.5 ± 0.1 (12 weeks); 0.3 ± 0.1(24 weeks)		
		Empty control	10 (6 weeks); 10 (12 weeks); 10 (24 weeks)	17.9 ± 0.4 (6 weeks); 17.7 ± 0.3 (12 weeks); 17.4 ± 0.3 (24 weeks)	2.3 ± 0.3 (6 weeks); 2.0 ± 0.1 (12 weeks); 1.7 ± 0.1 (24 weeks)	2.3 ± 0.3 (6 weeks); 2.0 ± 0.1 (12 weeks); 1.7 ± 0.1 (24 weeks)		
Løken et al., 2008 [23]	Modified O’Driscoll score (higher is better): 0 = not bonded, 1 = bonded at one end, 2 = bonded at both ends	Scaffold + MSCs	11	NA	0.91 (SD = 0.66)	Hyaline cartilage score: 1.45 (SD = 0.47); Necrosis: 1.45 (SD = 0.56); Chondrocyte clustering: 0.86 (SD = 0.51)	NA	Hyaline cartilage: *p* = 0.06 Necrosis: *p* = 0.09 Chondrocyte clustering: *p* = 0.03 for scaffold + MSC vs empty scaffold
		Empty Scaffold	11	NA	0.50 (SD = 0.63)	Hyaline Cartilage score: 1.05 (SD = 0.47); Necrosis: 1.05 (SD = 0.52); Chondrocyte clustering: 0.36 (SD = 0.39)	NA	
Zhou et al., 2008 [29]	Integration with adjacent cartilage, higher score means better outcome (Both edges integrated = 2; One edge integrated = 1; Neither edge integrated = 0)	PGA-HA + MSCs	10 (16 weeks); 10 (32 weeks)	14.2 ± 1.4 (16 weeks); 15.1 ± 1.4 (32 weeks)	1.7 ± 0.5 (16 weeks); 1.8 ± 0.4 (32 weeks)	3.5 ± 0.7 (16 weeks); 3.7 ± 0.5 (32 weeks)	2.3 ± 0.5 (16 weeks); 2.5 ± 0.5 (32 weeks)	Total: *p* < 0.05 between the MSCs-PGA-HA group and the two control groups
		Cell-free PGA-HA		6.8 ± 1.1 (16 weeks); 8.4 ± 1.7 (32 weeks)	0.9 ± 0.7 (16 weeks); 1.2 ± 0.8 (32 weeks)	2.1 ± 0.7 (16 weeks); 2.2 ± 0.6 (32 weeks)	1.4 ± 0.8 (16 weeks); 1.3 ± 0.9 (32 weeks)	
		Empty control	11	6.6 ± 2.5 (16 weeks); 7.4 ± 2.0 (32 weeks)	0.7 ± 0.8 (16 weeks); 1.0 ± 0.8 (32 weeks)	1.9 ± 0.7 (16 weeks); 2.0 ± 0.7 (32 weeks)	1.2 ± 0.6 (16 weeks); 1.0 ± 0.7 (32 weeks)	
Li et al., 2010 [30]	O’Driscoll score (lower is better): 0 = both edges integrated; 1 = one edge integrated; 2 = neither edge integrated	Bone-marrow derived MSCs	6 (12 weeks)	4.13 ± 0.83	0.25 ± 0.46	1.25 ± 0.46	0.38 ± 0.52	Total: *p* < 0.05 between BMSC group and all others. *p* < 0.05 between periosteum-, synovium-, adipose- and muscle-derived MSCs to control Integration: *p* < 0.05 between BMSC group and all others *p* < 0.05 between periosteum-, synovium- and adipose-derived MSCs and control Cell Morphology and Organization: *p* < 0.05 between BMSC group and all others *p* < 0.05 between synovium- and adipose-derived MSCs and control
		Periosteum-derived MSCs	6 (12 weeks)	12.0 ± 1.60	1.25 ± 0.46	2.50 ± 0.53	1.26 ± 0.71	
		Synovium-derived MSCs	6 (12 weeks)	11.4 ± 3.08	1.20 ± 0.72	2.67 ± 0.56	1.34 ± 0.42	
		Adipose-derived MSCs	6 (12 weeks)	10.98 ± 2.14	1.57 ± 0.42	2.05 ± 1.45	1.56 ± 0.58	
		Muscle-derived MSCs	6 (12 weeks)	12.23 ± 4.65	1.78 ± 0.80	1.98 ± 0.76	1.12 ± 0.34	
		Empty control	6 (12 weeks)	15.87 ± 0.64	2.00 ± 0.00	2.75 ± 0.46	1.87 ± 0.35	
Xie et al., 2010 [31]	Two integration scores for lateral and medial integration to adjacent cartilage (Bonded = 2, Partially bonded = 1, Not bonded = 0)	PLCL + MSCs	4 (3 months); 4 (6 months)	13.5 ± 2.1 (3 months); 21.7 ± 6 4.5 (6 months)	Medially: 0.4 ± 0.3 (3 months); 1.3 ± 0.3 (6 months); Laterally: 0.5 ± 0.4 (3 months); (6 months) 1.4 ± 0.5	4.1 ± 1.4 (3 months); 6.9 ± 2.5 (6 months)	NA	Total: *p* < 0.05 for PLCL + MSC vs cell-free PLCL and empty control at 3 months; *p* < 0.05 for MSC group at 6 months vs MSC group at 3 months. Empty control group was significantly worse at 6 months vs 3 months Hyaline cartilage: *p* < 0.05 for MSC group relative to other two groups at 3 and 6 months
		Cell-free PLCL	4 (3 months); 4 (6 months)	9.2 ± 2.6 (3 months); 12.3 ± 2.4 (6 months)	Medially: 0.3 ± 0.1 (3 months); 1.1 ± 0.2 (6 months); Laterally: 0.2 ± 0.2 (3 months); (6 months) 1.3 ± 0.6	1.0 ± 0.8 (3 months); 1.2 ± 0.4 (6 months)	NA	
		Empty control	4 (3 months); 4 (6 months)	11.6 ± 3.1 (3 months); 5.9 ± 3.2 (6 months)	Medially: 0.8 ± 0.3 (3 months); 0.6 ± 0.5 (6 months); Laterally: 0.7 ± 0.1 (3 months); 0.2 ± 0.2 (6 months)	2.1 ± 1.1 (3 months); 0.9 ± 0.2 (6 months)	NA	
Goodrich et al., 2016 [32]	Bonding to adjacent cartilage, (higher is better): Bonded at both ends of the graft = 2, Bonded at 1 end or partially at both ends = 1, Not bonded = 0	APEF + MSCs	12 (12 months)	NA	1.83 ± 0.39	Cellular morphology = 1.42 ± 0.79; Chondrocyte Clustering = 1.25 ± 0.62	Safranin-O staining = 1.33 ± 0.49	None significant
		Cell-free APEF	12 (12 months)	NA	2.0 ± 0.0	Cellular morphology = 1.25 ± 0.75; Chondrocyte Clustering = 1.25 ± 0.75	Safranin-O staining = 0.83 ± 0.71	
Wu et al., 2020 [33]	Modified O’Driscoll score, higher score means better outcome (Bonded at both ends of graft = 2, Bonded at one end = 1, Not bonded = 0)	Mixed pre- and undifferentiated MSCs	4 (7 months)	NA	1.53 ± 1.21	NA	NA	Integration: bilayered differentiated and mixed-phenotype groups significantly better compared to control (*p* < 0.05); bilayered differentiated group showed significant increase vs undifferentiated group
		Bilayered pre-differentiated MSCs	4 (7 months)	NA	2.0 ± 1.0	NA	NA	
		Undifferentiated MSCs	4 (7 months)	NA	1.0 ± 0.63	NA	NA	
		Empty control	4 (7 months)	NA	0.5 ± 0.63	NA	NA	

**Table 3 ijms-24-03227-t003:** PICOS inclusion and exclusion criteria for study selection.

Domain	Inclusion Criteria	Exclusion Criteria
Population	Animal subjects with surgically-created focal chondral defects of the knee joint.	Studies involving human or cadaveric subjects. Studies involving animals with diffuse osteoarthritis models. Ex vivo, in vitro, or in silico studies.
Intervention	Studies using implants consisting of autologous or allogeneic mesenchymal stem cells seeded into an organic scaffold or gel which were surgically introduced into a focal chondral defect.	Studies involving cell-free therapy, cell therapy without open surgical implantation, cells which are not bone marrow-derived, or other cell types such as chondrocytes, except as comparators. Studies using xenogeneic stem cells.
Comparison	Studies comparing the use of scaffolds/gel implants to cell-free scaffolds/gels, cell therapy without implantation, or empty controls.	None.
Outcome	Studies which quantitatively report the quality of integration of the regenerated and native cartilage, assessed histologically.	Studies reporting outcomes of joints other than the knee. Studies without a quantitative scoring system. Studies not involving a histological assessment. Studies which do not specify integration scores.
Study Type	English articles with full text available. Sample size greater than 10 animals.	Case reports, case series with fewer than 10 subjects, review articles.

## Data Availability

The data are contained within the article and Appendix A.

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
