# Peer review of "Bone Marrow-Derived Mesenchymal Stem Cell Implants for the Treatment of Focal Chondral Defects of the Knee in Animal Models: A Systematic Review and Meta-Analysis"

_ijms, 2023, doi:10.3390/ijms24043227_

Round 1

Reviewer 1 Report

The manuscript by Lee et al reports a systematic review and meta-analysis of the application of MSC to treat chondral defects. The outcome measures focus on the cartilage integration and the histological scores. The conclusion if this review is MSC is a promising source for cartilage integration and cartilage quality for treating chondral defects.

This is a thoroughly done study and I do not have anything to criticize. It was an exciting read and I find it very useful. This study will provide the baseline information for many researchers with a goal of clinical cartilage regeneration. To make the review more comprehensive I suggest incorporating the following edits.

Please discuss the functional assessment of the animals. Although the focus of this study has been integration and quality of cartilage, it won’t be hard to find how the animals functioned until the point of sacrifice. The specifics can be gait characteristics and pain level. I am not sure if this information is available in those papers but if it is existing – please summarize.

If any of the studies used chondrocytes as a control, discuss that. If not, explicitly state that. The comparison with cell-free control is applauded and indeed required for making a point about the usefulness of MSC but readers might want to know whether MSC is superior to chondrocytes for integration and making hyaline cartilage.

Discuss on the human studies with bone marrow derived stem cells, which were assessed with MRI. Although they were not included in the review, getting a summary in that line will be motivating. 

Author Response

Dear Reviewer,

Thank you very much for taking the time to review our work and provide your suggestions for improvements. We are pleased that you found the paper to be a valuable contribution to the current literature regarding cartilage repair techniques.

In accordance with your suggestions, we have amended the Discussion section of our manuscript as follows:

  1. We have explicitly stated that none of the included studies compared ACI to MSCs, as you requested. We have, however, included a discussion of human studies comparing MSCs to other interventions later in the manuscript.
  2. Following your suggestion to comment on functional outcomes of the animal subjects, we have reported the limited data available from the included papers. To elaborate on the assessment of clinical performance, we have summarized clinical findings from human studies using bone marrow-derived mesenchymal stem cells.
  3. Finally, we have written an additional paragraph reporting MRI outcome of humans treated with BMSCs, concluding that MRI findings are encouraging but not yet widely reported in the literature.

Thank you, once again, for your valuable comments. We believe these have greatly improved our submission.

Reviewer 2 Report

Lee E. et al. proposed a review and meta-analysis entitled "Bone Marrow-Derived Mesenchymal Stem Cell Implants for the Treatment of Focal Chondral Defects of the Knee in Animal Models: A Systematic Review and Meta-Analysis" for publication in International Journal of Molecular Sciences.

The same research group published in 2022 a review and meta-analysis entitled "The Use of Autologous Chondrocyte and Mesenchymal Stem Cell Implants for the Treatment of Focal Chondral Defects in Human Knee Joints—A Systematic Review and Meta-Analysis." in the same journal.

In that previous systematic review, most of the studies selected for meta-analysis investigated the use of autologous cartilage implant (ACI) and the authors focused their analysis on the degree of integration between the repair cartilage formed from the ACI and the native cartilage surrounding the defect. The outcomes of clinical scoring systems were also analyzed.  

The authors concluded that autologous chondrocytes-seeded scaffold implantation allowed to achieve an integration of good quality that was associated with clinical improvements.

However, Lee E. et al noticed that in human studies, morphology of the repair tissue and integration between the implant and native cartilage were both mainly assessed using magnetic resonance imaging as a non-invasive surrogate technique and did therefore not include quantitative and qualitative histological assessments of the repair tissue.

 As the previous work of Lee E. et al identified a paucity in studies using MSC therapy in humans, the research group now conducted a systematic review and meta-analysis of studies using bone marrow-derived MSC (BMSC)-seeded implants in animal models of focal cartilage defects of the knee to gain a comprehensive understanding of the ability of those cells to generate repair cartilage. The degree of integrative repair produced by BMSC, considering the quality of the repair tissue formed, was also investigated.

 The review was conducted in accordance with the PRISMA guidelines. Five databases were used for the selection of studies and quantitative results from histological assessments of integration quality as well as repair cartilage morphology and staining characteristics were extracted.

The meta-analysis demonstrated that high quality integration with the surrounding native cartilage was achieved using BMSC-seeded implants when compared to that of cell-free comparators or control groups.

Implant integration was associated with repair tissue morphology and staining properties looking like those of native cartilage. The high-quality integration scores observed between repair and native cartilage suggest that those implants could ameliorate the longevity of the implant.

 The authors thus proposed that the clinical potential of BMSC-seeded implant therapy should now be assessed in human patients as high-quality integration between regenerated and native cartilage has proven difficult to achieve, and yet it is vital for the lifespan of repair.

Author Response

Dear Reviewer,

We thank you for providing a comprehensive summary of our work, as well as your thorough understanding of the context in which it was written.

As no suggestions for revisions have been made, we have not amended the work following these comments.

Thank you very much for taking the time to review our manuscript.